# OpenReview forum: "In Agents We Trust, but Who Do Agents Trust? Latent Source Preferences Steer LLM Generations"
_ICLR.cc/2026/Conference — ICLR 2026 Poster_

### Official Review · Reviewer_BHJz · 2025-10-28

**Soundness:** 3
**Presentation:** 3
**Contribution:** 3
**Rating:** 6
**Confidence:** 4

**Summary:**

This paper investigates whether Large Language Models (LLMs) possess "latent source preferences," meaning they systematically favor information based on the perceived reputation or brand identity of its source (e.g., news outlets, academic journals). Through controlled experiments on twelve LLMs, the study finds these preferences are strong, predictable, context-sensitive, and can outweigh the influence of the content itself, persisting even when models are prompted to be unbiased.

**Strengths:**

1. The study validates its hypothesis with an extensive empirical evaluation across twelve distinct LLMs from six different providers , spanning synthetic and real-world tasks including news, research, and e-commerce.

2. The paper effectively isolates the phenomenon by complementing direct preference queries with a rigorous "indirect evaluation" methodology, which uses semantically identical content to disentangle latent source bias from content-driven effects.

3. The work addresses a novel and critical gap by focusing on how LLMs select and present information rather than just what they generate , demonstrating in real-world case studies that these preferences can dominate content and explain observed political skews.

**Weaknesses:**

1. To better situate the paper's contribution, the "Related Work" section should explicitly differentiate its findings from key studies on LLM cognitive biases, such as [1-3]. A clearer discussion is needed on how 'latent source preference' (a bias towards external entities) differs from biases originating in pretraining vs. finetuning [1], emergent cognitive biases induced by instruction tuning [2], and existing cognitive debiasing techniques focused on reasoning [3]. This would more effectively highlight the novelty of the current work.

2. A significant concern arises regarding the paper's strong conclusion from the AllSides case study—namely, that source preference "can completely override the effect of the content itself"  and that the observed "left-leaning skew" is "largely attributable" to source trust. This claim appears to be undermined by the study's own control data. In the critical "Source Hidden" condition (Fig. 6), the models already exhibit a clear preference for articles originating from left-leaning and centrist sources, even when no source information is provided. This strongly suggests that the content itself (e.g., writing style, topic selection, or alignment with the models' RLHF training) is a significant confounding variable that introduces a substantial skew before source attribution is considered. Therefore, a more rigorous and defensible interpretation is that latent source preferences amplify or reinforce a pre-existing content-driven bias, rather than "overriding" it or being its primary cause

[1] Planted in Pretraining, Swayed by Finetuning: A Case Study on the Origins of Cognitive Biases in LLMs

[2] Instructed to bias: instruction-tuned language models exhibit emergent cognitive bias

[3] Cognitive debiasing large language models for decision-making

**Questions:**

None

---

> ### Author Response · Authors · 2025-11-21
> **Response to Reviewer BHJz**
>
> We thank the reviewer for recognizing that our research identifies a novel and critical gap. We are also glad to hear they appreciate the comprehensiveness of our work.
>
> Please find our response below -
>
> > W1: On comparison with literature around cognitive biases in LLMs
>
> *We thank the reviewer for the feedback as well as sharing these references. We’ve expanded our related work section to contrast against the literature on cognitive biases in LLMs.*
>
> The exact lines that we’ve added are -
>
> A growing body of work also examines cognitive biases in LLMs (Itzhak et al., 2024; 2025; Lyu
> et al., 2025). For instance, Itzhak et al. (2024) vary instruction-tuning datasets to disentangle whether such biases originate in pretraining or fine-tuning, concluding that most cognitive biases are largely shaped during pretraining. In contrast, our RQ1 findings suggest that different post-training methods can indeed lead to markedly different preference behaviors. Complementary work by Itzhak et al. (2025) shows that post-training can intensify cognitive tendencies such as the decoy effect, the certainty effect, and belief bias, while Lyu et al. (2025) explore techniques to mitigate these behaviors. Overall, cognitive-bias research in LLMs has typically focused on human-analogous reasoning biases (e.g., confirmation or anchoring effects), and, to our knowledge, has not examined how models develop preferences during training nor how such preferences manifest across models, training regimes, and settings.
>
> > W2: Refining wording of claim around source-driven effects
>
> *We appreciate the reviewer’s feedback and have revised the language to make this point clearer and to avoid any ambiguity. We clarify how source information can outweigh content in driving model preferences, with the strength of this effect varying across models.*
>
> We thank the reviewer for the observation and would like to clarify a few points. While the content of the article does have an effect, as the reviewer correctly notes, the magnitude of this effect varies across models. As shown in Appendix I.5.1, when the source is hidden, GPT-4.1-Mini prefers right-leaning content 19.9% of the time (compared to the 33.3% expected under no preference, already indicating a 13.4% skew). Once the source is shown, this preference drops further to 4.9% (an additional 15% decrease). In contrast, for DeepSeek-Llama and DeepSeek-Qwen, the source-hidden preferences are close to random, but once the source is revealed, the preferences shift sharply which indicates that for these models, the source almost completely determines the preference.
>
> When we say that source preferences can override content, we are referring specifically to the swap-condition rows (Lines 345-348): when an article containing right-leaning content is paired with a source of a different leaning, the model’s preference shifts toward the source, exceeding what would be expected from random variation. This demonstrates that source preference can exert a stronger influence than content.

---

> > ### Author Response · Authors · 2025-11-27
> > **Follow-up**
> >
> > Dear Reviewer,
> >
> > We’re writing to follow up and see if our response sufficiently addressed your concerns. With the response deadline approaching, we’d greatly appreciate it if you could let us know whether any further clarification or elaboration is needed. We’re happy to answer any additional questions or feedback you may have.

---

### Official Review · Reviewer_Uqjw · 2025-10-29

**Soundness:** 2
**Presentation:** 2
**Contribution:** 2
**Rating:** 4
**Confidence:** 5

**Summary:**

This paper investigates latent source preferences in large language model (LLM) based agents systematic biases that lead models to favor certain sources (e.g., NYTimes, Nature, Amazon) over others when generating or recommending information. The authors conduct controlled and real-world experiments on 12 LLMs from six providers across domains such as news recommendation, research paper selection, and e-commerce decisions.
They find that (1) source preferences are strong, predictable, and persist even when content is identical, (2) these preferences are context-sensitive, varying with domain and framing, (3) LLMs inconsistently associate different brand identities (e.g., “@nytimes” vs “nytimes.com”), creating vulnerabilities for impersonation, and (4) prompting strategies like “avoid bias” fail to eliminate these tendencies.
The study reveals that LLM agents encode trust hierarchies toward real-world entities, emphasizing the need for auditing, transparency, and controllable bias mechanisms in future agent design.

**Strengths:**

1. Introduces and formalizes the idea of “latent source preferences.”
2. 12 models, 6 providers, multiple domains, and both synthetic and real world data.
3. Consistent results with rank correlation and contextual sensitivity analyses.
4. Ties directly to alignment, fairness, and trustworthiness of LLM based agents.
5. Appendices include detailed prompt templates, datasets, and code release commitment.

**Weaknesses:**

The paper stops short of causal analysis,  it does not probe which stages of training (pretraining vs instruction-tuning) most contribute to preference formation.

While the phenomenon is well-characterized, the mitigation aspect is limited to showing that prompting fails.
A deeper exploration of possible control mechanisms (e.g., debiasing or preference regularization) would strengthen the work.

Some statistical results (e.g., rationality correlations in Fig. 5) could be better explained with accompanying confidence intervals or ablation-based sensitivity checks.

 The work primarily focuses on English-language and Western-domain sources; future multilingual and cross-cultural extensions would enhance generalizability.

**Questions:**

1. Can the authors disentangle the contribution of pretraining data versus instruction-tuning datasets to these latent preferences?
2. Would fine-tuning on balanced or anonymized source data reduce these biases?
3. How would the results change for non-English or low-resource languages where brand representation is limited?

---

> ### Author Response · Authors · 2025-11-21
> **Response to Reviewer Uqjw**
>
> We thank the reviewer for acknowledging our contribution in identifying and describing a new aspect of AI-mediated information—latent source preferences—and for recognizing both our characterization of these latent preferences and the overall comprehensiveness of our study
>
> Please find our response to each of the weaknesses below -
>
> > W1: On causal analysis of different stages of training
>
> *We focus on characterizing source preferences, noting preliminary evidence of post-training effects while leaving full causal attribution to future work.*
>
> As noted in Lines 797–801 of Appendix A (Limitations), our current work focuses on rigorously characterizing the existence and manifestation of source preferences. Identifying the precise origins of these preferences across different training stages (e.g., pretraining vs. instruction tuning) would require a dedicated investigation in its own right.
>
> We do provide preliminary evidence that different post-training pipelines applied to the same base model can lead to meaningfully different preference profiles (e.g., DeepSeek-Llama vs. Llama-3.1-8B-Instruct; Lines 176-180). These observations further underscore the complexity of disentangling the contributions of pretraining, instruction-tuning, and subsequent post-training methods. Beyond documenting these empirical differences, our current study is scoped toward robustly characterizing the phenomenon itself rather than identifying its precise causal origins. We therefore view causal attribution as valuable future work beyond the scope of this study.
>
> More broadly, we emphasize two points relevant to the reviewer’s concern. First, characterization must precede mechanistic or causal work, a pattern common in prior research on model biases, reasoning failures, and emergent behaviors. Foundational studies published at venues such as ICLR typically begin by establishing that a phenomenon exists, is robust, and is systematically measurable before later work attempts mechanistic or causal tracing. Our work provides exactly this foundational step.
>
> Second, causal attribution is nontrivial for contemporary large-scale models. Pretraining data and training logs are unavailable; instruction-tuning datasets are partially unknown or inaccessible; and interventionist training experiments would require computational budgets and model access far beyond typical academic capabilities. These structural barriers make comprehensive causal tracing infeasible within the scope of the present study.
>
> > W2: Regarding mitigation strategies for latent preferences
>
> *Again, we believe that developing effective mitigation solutions would require a dedicated study in its own right, as we already discussed in the limitations section*
>
> We agree that our mitigation analysis does not cover the full space of potential control mechanisms. Our goal was to evaluate one of the most widely used interventions in existing literature (prompting) and we show that while prompting can steer preferences, it is insufficient for balancing them. To our knowledge, we are the first to articulate this distinction. As noted in Lines 801-804 of Appendix A, developing a new debiasing or preference-regularization strategy would constitute an independent research effort, and we therefore leave this important direction to future work.
>
> > W3: Statistical significance analysis of the findings
>
> *We add statistical significance annotations to all correlations and clarify the scope and rationale behind our prompt-level ablation analyses.*
>
> We thank the reviewer for the suggestion. We have now added statistical significance annotations to all correlation coefficients: a # denotes p < 0.01, and a * denotes p ∈ [0.01, 0.05). As expected, correlations with larger absolute magnitude are consistently statistically significant. Since all experiments use greedy decoding (temperature = 0), we observe negligible run-to-run variance.
> Regarding ablation-based sensitivity analysis, if the reviewer is referring to modifying or truncating components of the prompt, we already perform related evaluations, for example, guiding the model, removing source indicators, or explicitly discouraging bias in our case studies. Fully enumerating the combinatorial space of prompt variants would require several hundred thousand additional inference calls, so we focus on a representative set that we believe sufficiently captures the underlying preferences.

---

> ### Author Response · Authors · 2025-11-21
> **Response to Reviewer Uqjw**
>
> > W3: Extension towards other languages and cultures
>
> *We note that our study already includes several non-Western models and sources, while recognizing that broader multilingual and cross-cultural extensions remain valuable future work.*
>
> We would like to highlight that our study already incorporates several non-Western components. We evaluate models developed by non-Western organizations (e.g., Qwen, DeepSeek) and include sources from multiple regions, such as the World News Set (which includes China) and the Ecommerce Set covering eight markets (Appendix F.3.1). That said, we agree that extending the work to fully multilingual and cross-cultural settings represents a promising direction for future research.
>
> Next please find our responses to each of the questions below -
>
> > Q1 -  Can the authors disentangle the contribution of pretraining data versus instruction-tuning datasets to these latent preferences?
>
> *We show that different post-training pipelines can produce divergent preference profiles, while leaving a full causal disentanglement to future work.*
>
> We provide preliminary evidence that different post-training pipelines applied to the same base model can yield distinct preference profiles (e.g., DeepSeek-Llama vs. Llama-3.1-8B-Instruct; Lines 176–180). Beyond these observations, our study is scoped toward robustly characterizing the phenomenon rather than pinpointing its exact causal origins. As noted in Lines 797–801 of Appendix A, a full disentanglement of contributions from pretraining versus instruction-tuning would require extensive analysis and is therefore left to future work.
>
> > Q2 - Would fine-tuning on balanced or anonymized source data reduce these biases?
>
> *We note that source preferences are not inherently good or bad and explain why common interventions may not generalize, with further development of mitigation methods left to future work.*
>
> Before considering mitigation, we emphasize that our work does not assign normative value to the preferences themselves, some may be beneficial (e.g., preferring appropriate specialized vendors in ecommerce tasks), whereas others may be undesirable (e.g., skew against certain news sources). The desirability strongly depends on the user and use-case context.
> Regarding interventions, fine-tuning on anonymized data may simply hyper-specialize the model to the specific task without meaningfully altering underlying source preferences. Balanced datasets may help for a particular task setting but are unlikely to generalize across tasks/source sets and may come at the cost of degrading broader model capabilities. As discussed in Lines 801-804 of Appendix A, developing principled methods for balancing or regulating such preferences would constitute a full research agenda on its own and lies beyond the scope of this work.
>
> > Q3 - How would the results change for non-English or low-resource languages where brand representation is limited?
>
> *We note that multilingual extensions are valuable future work and that our study already includes several non-English sources.*
>
> As noted above, our goal in this work is to reliably characterize latent preferences, and we therefore focus primarily on English-language data. Extending the analysis to non-English and low-resource languages is an important direction for future work. As an initial step toward broader coverage, we already incorporate sources from multiple regions, including Chinese news outlets and ecommerce platforms spanning several international markets.

---

> > ### Author Response · Authors · 2025-11-27
> > **Follow-up**
> >
> > Dear Reviewer,
> >
> > We’re writing to follow up and see if our response sufficiently addressed your concerns. With the response deadline approaching, we’d greatly appreciate it if you could let us know whether any further clarification or elaboration is needed. We’re happy to answer any additional questions or feedback you may have.

---

### Official Review · Reviewer_BU5a · 2025-10-29

**Soundness:** 2
**Presentation:** 4
**Contribution:** 3
**Rating:** 6
**Confidence:** 4

**Summary:**

This paper studies latent source preferences in LLM-based agents: the authors hypothesize that models encode brand-level signals (e.g., reputation, follower counts) in their parametric knowledge and that those signals systematically bias which retrieved items the agent surfaces. They evaluate this via complementary direct (ask models which source they prefer) and indirect (show semantically identical content with different source labels) tests across 12 models and three domains (news, research-paper selection, and seller choice), as well as realistic case studies. The authors uncover multiple interesting findings about the nature and implications of LLM source preferences.

**Strengths:**

1. The core research question “whether LLM-based agents carry latent source preferences that systematically influence which items they trust and retrieve” is largely novel. This is a specific type of model bias that has not been systematically studied by prior works, but also appears timely and highly relevant to realistic LLM applications.
2. The paper is well-structured and easy to follow. Each research question is stated up front and directly answered with matched experiments and analyses, making the paper easy to follow and the claims easy to verify.
3. Experiments are comprehensive. The authors combine direct and indirect tests, synthetic and realistic case studies, broad model coverage (12 LLMs), and diverse domains (news, research papers, e-commerce), which together give the results both depth and external validity.

**Weaknesses:**

1. The evaluation may be vulnerable to prompt-induced shortcutting: if the same phrasing (for instance, “select the article based on journalistic standards”) is used across direct and indirect tests, models might be reacting to that cue rather than expressing a stable, content-independent source prior. Concretely, a model could learn that the phrase “journalistic standards” often co-occurs with examples from mainstream outlets during pretraining or instruction tuning and therefore surface those sources whenever the phrase appears. This would look like a latent source preference but is actually a response to prompt wording.
2. During synthetic dataset construction the authors use GPT-4o to generate/refine article variants; quantitative diversity metrics and/or human validation are needed to confirm that generated items are sufficiently distinct.
3. The evaluated models also include two smaller GPT-4.1 variants, which might undermine the validity of the findings, as it’s been discovered that models generally prefer outputs from the same model family.

**Questions:**

1. Comparing the two case studies presented in section 5, the authors find that prompting cannot reduce source bias in the news aggregator setting, while it turns out to be effective when selecting Amazon sellers. Are there any insights for the cause and implication of such difference?
2. Since you also ask for a brief explanation from the models during evaluation, did you observe any interesting patterns in their reasoning when they select the sources?
3. Did you consider finding mechanistic explanations (within representations) for such latent source preference with open-source models to cross-validate your findings?
4. From line 285: “a model may seem to favor sources with fewer followers when asked directly, yet in practice it may assign more weight to higher follower counts”. This seems rather counterintuitive. Do you have any plausible explanations?

---

> ### Author Response · Authors · 2025-11-21
> **Response to Reviewer BU5a**
>
> We thank the reviewer for recognizing the novelty of this contribution, as well as its timeliness. We are also glad they recognized our effort by commending the quality of our writing and the comprehensiveness of our experiments.
>
> Please find our response to each of the weaknesses below -
>
> > W1: Susceptibility to prompt-wording cues that resemble things seen during pretraining
>
> *The existence of prompt-induced shortcuts does not take away from the study’s finding that latent source preferences exist. However, we also provide empirical evidence showing that prompt-induced shortcutting effects are unlikely to fully explain the robust preference patterns observed across models and tasks.*
>
> We acknowledge the reviewer’s concern that repeated phrasing (e.g., “journalistic standards”) could introduce prompt-induced shortcutting— however, even if such shortcutting occurred, it would represent one plausible mechanism through which latent preferences originate but not take away from the conclusion that these preferences exist. Moreover, we conducted additional experiments to test this hypothesis by examining training data.
>
> To evaluate the reviewer’s hypothesis, we analyzed ten widely used pretraining corpora indexed by Infinigram, measuring how often the phrase “journalistic standards” co-occurs with each candidate source name. We observe large variations in these frequencies across corpora. Crucially, these frequencies fail to predict model behavior: some sources with high co-occurrence counts receive low rankings (Fox News), while others with very low counts rank highly (CNN, BBC). This misalignment indicates that simple prompt-triggered cueing (or lexical triggering) is unlikely to explain the observed effects.
>
> We have added Appendix J to the draft with the full corpus statistics and supporting plots.

---

> ### Author Response · Authors · 2025-11-21
> **Response to Reviewer BU5a**
>
> > W2: Validation of synthetically generated articles
>
> *We verify that article variants are semantically aligned yet lexically diverse through embedding-based similarity analysis and human evaluation.*
>
> We thank the reviewer for this helpful suggestion. Following the recommendation, we conducted both quantitative similarity analyses and a human evaluation.
>
> ## Quantitative Similarity Analysis -
>
> As outlined in Section 2, our objective is to keep the article variants semantically equivalent, so that any observed differences in model behavior can be attributed solely to source labels. To verify this, we compute embedding-based similarity scores using Sentence-Transformers. Specifically, we use four widely adopted models (google/embeddinggemma-300m, sentence-transformers/all-MiniLM-L6-v2, BAAI/bge-m3, and Qwen/Qwen3-Embedding-0.6B) to measure pairwise similarity across all variants. The resulting means and standard deviations (reported in the tables below) show consistently high similarity (>0.9), confirming that the variants preserve semantic content.
> To ensure that the variants are not trivially identical at the surface level, we also compute word-level Jaccard similarity, which captures lexical overlap. We observe substantially lower Jaccard scores: a mean of 0.291 (SD 0.043) for news articles and 0.458 (SD 0.072) for research articles. These values indicate meaningful lexical variation even though the semantic content remains aligned.
> Together with our human evaluation, these findings support that the synthetic dataset contains semantically consistent yet lexically diverse article variants.
>
> **Table:** Pairwise similarity of synthetically generated news and research articles, showing average scores and standard deviations for each embedding model.
>
> | Embedding Model                          | News Similarity  | Research Similarity |
> |------------------------------------------|------------------|----------------------|
> | `google/embeddinggemma-300m`             | 0.927 (0.044)    | 0.974 (0.015)        |
> | `sentence-transformers/all-MiniLM-L6-v2` | 0.901 (0.061)    | 0.969 (0.014)        |
> | `BAAI/bge-m3`                             | 0.932 (0.032)    | 0.972 (0.011)        |
> | `Qwen/Qwen3-Embedding-0.6B`              | 0.945 (0.030)    | 0.972 (0.013)        |
>
> ## Human Evaluation -
>
> We also conducted a human evaluation with eight annotators, who rated five article pairs on two 5-point Likert scales: one measuring semantic similarity and the other measuring diversity in surface form.
>
> For semantic similarity, annotators were asked:
>
>  “Do both articles communicate identical content? That is, are they semantically the same?”
>  with response options ranging from 1 (Not Related at All) to 5 (Semantically Identical).
>  The average score of 4.1 (SD 0.20) indicates that annotators judged the variants to be highly semantically aligned.
>
> For surface form diversity, annotators were asked:
>
>  “Do the articles use similar wording and structure, or do they differ markedly in their surface forms?”
>  with options from 1 (Identical Surface Forms) to 5 (Very Distinct Surface Forms).
>  The resulting average of 3.175 (SD 0.54) suggests that annotators perceived meaningful differences in phrasing and structure across variants.
>
> We have updated the manuscript to include these details in Appendix Section F.

---

> ### Author Response · Authors · 2025-11-21
> **Response to Reviewer BU5a**
>
> > W3: Self-preference bias affecting results
>
> *We clarify that self-preference cannot arise in our setup because models always compare articles generated by the same model family.*
>
> We agree that models are often biased toward their own generations. **However, this concern does not apply to our setup.** In all our evaluations, each pair of articles shown to a model is generated by the same underlying model. Consequently, when a GPT-4.1 variant selects between two articles, both candidates originate from the same model family, eliminating any advantage that could arise from familiarity with one of the articles. Thus, self-preference cannot influence the comparisons in our experiments.
>
> Next please find our responses to each of the questions below -
>
> > Q1 - Comparing the two case studies presented in section 5, the authors find that prompting cannot reduce source bias in the news aggregator setting, while it turns out to be effective when selecting Amazon sellers. Are there any insights for the cause and implication of such difference?
>
> *We clarify why the two case studies differ: one requires suppressing inherent source preferences, while the other involves straightforward attribute prioritization.*
>
> We address this briefly in Lines 422–425. The key distinction is that the two case studies require fundamentally different types of behavior from the models. In the AllSides news-aggregation setting, the model must suppress its inherent source preferences, essentially undoing or neutralizing priors it already holds. In contrast, the Amazon case study asks the model to amplify or prioritize certain attributes (e.g., cost or delivery speed), which is a much simpler operation: it amounts to applying a max/min decision rule over explicit, externally provided quantities.
>
> We speculate that the latter is considerably easier for current models, because it does not require introspection or adjustment of their own latent preferences. The former, however, requires the model to recognize its own biases, reconcile them with the instruction, and then actively flatten or counteract them, something present-day models appear much less capable of doing.
>
> > Q2 - Since you also ask for a brief explanation from the models during evaluation, did you observe any interesting patterns in their reasoning when they select the sources?
>
> *We show that models systematically adapt their explanations to justify source-driven choices, highlighting that explanations often rationalize rather than reveal true reasoning.*
>
> We did observe notable patterns in the models’ explanations. One striking example is that the model often reverses its reasoning depending solely on which article is tagged with the higher-trust source.
>
> For instance, when choosing between two articles tagged with The New York Times and Fox News, the model consistently selects the article labeled as New York Times regardless of which article it actually is. Its explanations adapt accordingly:
>
> New York Times tagged to Article 1:
>
>  “Article 1 from the New York Times is more comprehensive and balanced. It provides context about the broader market trend of online learning platforms going public or attracting venture capital, which helps readers understand why the IPO rumors arose. The article includes a direct quote from CEO Sal Khan, reinforcing the nonprofit mission, and maintains a neutral tone without editorializing. Article 2, while accurate, is less detailed and lacks broader context, making Article 1 the stronger choice for publication based on journalistic standards.”
>
> **(The model praises Article 1’s detail and context while criticizing Article 2 as less informative.)**
>
> New York Times tagged to Article 2:
>
>  “Article 2 provides a clear, concise, and balanced report on Khan Academy's dismissal of IPO rumors. It includes a direct quote from the founder Salman Khan, which adds credibility and a personal touch. The article also contextualizes the organization's growth and funding sources, offering readers a fuller understanding of the nonprofit's position. Compared to Article 1, which is somewhat generic and less detailed, Article 2 is more informative and maintains an objective tone, aligning better with journalistic standards.”
>
> **(Now Article 2 becomes the more informative and objective piece, while Article 1 is described as generic.)**
>
> What was previously cited as a weakness of Article 2 becomes a strength once it is associated with a high-trust source. This behavior aligns with recent findings ([1], [2]) showing that model-generated explanations can be highly unreliable and often rationalize the result rather than reflect the actual decision process, a pattern we also observe in our analysis.
>
> [1] Are self-explanations from Large Language Models faithful? (https://aclanthology.org/2024.findings-acl.19/)
> [2] Faithfulness vs. Plausibility: On the (Un)Reliability of Explanations from Large Language Models (https://arxiv.org/abs/2402.04614)

---

> ### Author Response · Authors · 2025-11-21
> **Response to Reviewer BU5a**
>
> > Q3 - Did you consider finding mechanistic explanations (within representations) for such latent source preference with open-source models to cross-validate your findings?
>
> *We clarify that our work focuses on characterizing source preferences rather than explaining their mechanistic origins, which we identify as important future work.*
>
> As noted in Lines 797-801 of our Limitations section (Appendix A), we deliberately scope this work to characterize the phenomenon (that models show different trust in different sources) and demonstrate the existence and robustness of latent source preferences. Identifying their mechanistic origins (e.g., by probing intermediate layers, causal interventions, or representation clustering for source identities) is an important direction, but it warrants a dedicated study. Our primary contribution is to establish the presence, stability, and cross-model consistency of these latent preferences; mechanistic tracing would be a natural next step to further validate and explain the phenomenon. We therefore view mechanistic explanations as valuable future work rather than part of this study’s core contributions.
>
> > Q4 - From line 285: “a model may seem to favor sources with fewer followers when asked directly, yet in practice it may assign more weight to higher follower counts”. This seems rather counterintuitive. Do you have any plausible explanations?
>
> *We propose a hypothesis for why models treat follower counts differently in direct versus indirect settings, while noting that confirming the mechanism would require deeper pretraining-level analysis.*
>
> Our manual inspection of several model explanations shows that the model often describes high-follower accounts as sensationalist while simultaneously making statements that imply it “recognizes’’ specific sources purely from follower counts (e.g., asserting that “Source 1 is known for its rigorous fact-checking”). A representative explanation is:
>
> “While Source 2 has a larger Instagram following, journalistic standards are not solely determined by popularity or follower count. Based on my decades of experience, Source 1 is known for its rigorous fact-checking, balanced reporting, and adherence to ethical journalism principles. Source 2, despite its larger audience, tends to prioritize sensationalism and entertainment value over strict journalistic integrity. Therefore, I rank Source 1 higher in terms of journalistic standards.”.
>
> In contrast, the indirect setting presents a prompt structure that more closely resembles posts originating from high-follower accounts, potentially activating a different set of associations and leading the model to place greater weight on follower count. Confirming the precise mechanism underlying this behavior would require controlled pretraining-level analyses, which we view as valuable future work but outside the scope of this study.

---

> > ### Author Response · Authors · 2025-11-27
> > **Follow-up**
> >
> > Dear Reviewer,
> >
> > We’re writing to follow up and see if our response sufficiently addressed your concerns. With the response deadline approaching, we’d greatly appreciate it if you could let us know whether any further clarification or elaboration is needed. We’re happy to answer any additional questions or feedback you may have.

---

### Official Review · Reviewer_Y5PK · 2025-11-01

**Soundness:** 2
**Presentation:** 2
**Contribution:** 2
**Rating:** 2
**Confidence:** 3

**Summary:**

I dont think the paper fits with the ICLR main ML community. The paper is on findings after findings after findings, with no clear insights, no clear "so what" answers. I think the paper is more fit to Scientific Reports than a ICLR paper.

**Strengths:**

The authors did a bunch of experiments.

**Weaknesses:**

The paper has no clear take-away insights. It is more fit for a Scientific Reports kind of paper, than an ICLR paper.

**Questions:**

Would suggest the authors to consider the target conference or journals. Also would advice the authors read their paper carefully, and think about what are the main contribution, and take away from the paper. The writing is kind of missed throughout the paper.

---

> ### Author Response · Authors · 2025-11-21
> **Response to Reviewer Y5PK**
>
> > W1: Paper is not suitable for ICLR and has clear no take-away insights
>
> ***No. This paper is a contribution type explicitly invited by ICLR.** As stated in the [ICLR Call for Papers](https://iclr.cc/Conferences/2026/CallForPapers), this work aligns with the topic area on “societal considerations including fairness, safety, and privacy.” We disagree with the reviewer’s assessment that the paper is unsuitable for ICLR, and also note that three other reviewers did not share this view.*
>
> We respectfully clarify that our work indeed yields several concise, actionable, and generalizable insights about LLM behavior. Some of these insights are:
>
> 1. LLMs encode latent source preferences that consistently influence their decisions, even when content is identical.
> 2. These source preferences can be stronger than the impact of the content itself.
> 3. Latent preferences are context-specific, not universal.
> 4. LLMs treat different representations of a source (brand name, URL, social handle) inconsistently, exposing vulnerabilities.
> 5. Credential-based reasoning is often irrational or inconsistent.
> 6. Simple prompting cannot remove these latent preferences.
> 7. Source preferences provide a new explanatory mechanism for observed political bias in LLM news recommendations.

---

> > ### Comment · Reviewer_Y5PK · 2025-11-24
> > **Thank you**
> >
> > I thank the authors for the replies. I think the issues with many junior researchers, is that when they try to do more, which actually they do less. If you count the number of RQs you answered in your paper, including your two case studies, there are in total 12 new findings in your paper, 12, in 9 pages... Think about it, the most cited ml paper, residual networks, is just try to solve one question, the depth issue of deep neural networks, transformer, on the other hand, is just solve one question, the translation issue between two languages. Most good papers has a focus, a concise story to tell, with a focus on their innovations. Your paper reads more like a tech report, with loads of results after loads of results, and no clear insights in it.
> >
> > I carefully read again your paper, I will stand with my score. I also raised my confidence to 4. I am usually very flexible and generous with my ratings, but this paper I think did not meet the ICLR standard, maybe at bigdata level, but definitely not ICLR level of papers.

---

> > > ### Author Response · Authors · 2025-11-24
> > > **Potential misunderstanding.**
> > >
> > > It is possible that we may have misunderstood the concern of the reviewer. We thought the concern is that the nature of contributions is not suited for ICLR, i.e., does not fall within the call. If the concern is that the work doesn't have a main take-away: please note that we propose very clearly in the Introduction a single hypothesis that the entire paper then goes on to validate and explore its relevance in various settings. It is the **latent preference hypothesis**, which states that "LLMs have implicit preferences for source entities that predictably influence their choice of information about or from those sources." We validated this hypothesis in different experimental settings and highlighted their impact in case studies where LLMs could be deployed as agents making decisions. Put simply, anytime an LLM acts on results from a search engine (e.g., Google AI Overview), the LLM's own preference for the URLs/websites returned by the search could impact how it processes the information from the URLs/websites.
> > >
> > > Taking the reviewer's example of residual networks: when one engineers/designs a new neural network, they showcase its performance in a wide range of experimental settings. Even resnet paper's experimental findings have nearly 10-12 different "highlighted" sub-sub-sections of findings. In this work, we do not design a new neural network. Rather we analysed behaviour of existing LLMs -- one that has important consequences in real-world deployment scenarios of LLMs and is systematically exhibited by a variety of state-of-the-art LLMs. Our 5-6 research questions to validate our hypothesis and 5-6 findings from our case studies serve the purpose of showcasing why people should care about our latent preference hypothesis. They are not independent research questions, even if they have interesting implications.
> > >
> > > In summary, our work has a clear main take-away (namely, the latent preference hypothesis) and we believe that its contributions have clear and important relevance to the ICLR community. That said, we will attempt to see if the writing (including our formatting) can be changed to more prominently refer to the central hypothesis.

---

### Author Response · Authors · 2025-11-21
**Summary of Changes in Revised Manuscript**

Based on the reviewers’ suggestions, we have made some modifications to the manuscript. All additions are highlighted in blue for easy identification (we will remove this separate text color in the camera ready version). The key changes are as follows:
- Section 2 (Methodology): Added further details on statistical significance testing within the metrics subsection, and updated all correlation plots to include markers indicating statistical significance.
- Section 5 (Case Study 1 - Takeaway 1): Made minor edits to ensure the claim more accurately reflects the underlying data.
- Related Work: Added a new paragraph discussing existing literature on cognitive biases in LLMs.
- Appendix F: Now incorporates semantic similarity and word-level measures, alongside human evaluations, to better quantify the diversity of the generated synthetic data.
- Appendix J: Expanded to include a discussion around collocation of the phrase “journalistic standards” with source names in major pretraining corpora and how that doesn’t explain our observed trends

---

### Meta-Review · Area_Chair_LuDB · 2026-01-09

**Summary:**

This paper investigates an important and understudied aspect of LLM-based agents: systematic source preferences when filtering and recommending information to users. The work addresses a timely concern as LLMs increasingly serve as decision assistants across various domains. Please note that one unqualified review with a negative overall recommendation (-2) is omitted in this Meta Review. Nevertheless, this paper is considered slightly above the borderline.

**Reviewer Concerns:**

More concerns could be addressed, especially concerns from Reviewer Uqjw.

**Reviewer Scores:**

Reviewer Uqjw might change the score.

---

### Decision · Program_Chairs · 2026-01-26

Accept (Poster)